



# Synoptic observation of full mesoscale eddy lifetime and its secondary instabilities in the Gulf of Mexico

Charly de Marez[1]

[1]Háskóli Íslands, Reykjavík, Ísland

**Correspondence:** Charly de Marez (charly@hi.is)

**Abstract.** Mesoscale eddies are crucial to ocean circulation, climate, and tracer transport. Yet, their full life cycle has never been observed synoptically at high resolution. In this study, we use novel SWOT satellite altimetry data to present the first synoptic characterization of a Loop Current Eddy's life cycle in the Gulf of Mexico, from formation to dissipation. SWOT allows for the direct observation of key dynamical processes—such as eddy shielding, high-mode instabilities, and dipolar interactions—
that were previously only described in theoretical and numerical studies. These observations challenge the traditional view of eddies as simple, elliptical structures, emphasizing the role of mesoscale interactions in their evolution. Furthermore, SWOT captures intense submesoscale turbulence at the eddy's rim, revealing secondary instabilities likely driving its decay. These findings not only validate decades of vortex theory but also offer new insights into oceanic turbulence dynamics.

## 1    Introduction

The study of oceanic vortices, or eddies, has been a cornerstone of oceanography for decades, driven by the critical role these features play in biological activities (Chelton et al., 2011a), tracer transport (Zhang et al., 2014), and properties of the water column (Dong et al., 2014). In particular, the mesoscale (10-100 km) eddy field is at least as energetic as the large scale circulation (Zhang et al., 2014), essential for the air-sea interactions (Small et al., 2008), and thus for the evolution of climate (Palter, 2015). Early observations of oceanic mesoscale eddies were limited by the resolution of available data, primarily relying
on ship-based measurements, which provided isolated snapshots of ocean conditions (Fuglister and Worthington, 1951). With the advent of satellite measurements in the late 20th century—first with Sea Surface Temperature (SST) radiometry (Richardson et al., 1973), then altimetric Sea Surface Height (SSH) (Chelton, 2001)— the tracking of mesoscale eddy surface signatures improved dramatically. This enabled global-scale analyses (Chelton et al., 2007), though at low resolution (typically 1/4°). In parallel, the theoretical and numerical understanding of mesoscale eddy dynamics has advanced significantly, offering valuable
insights into eddy behavior through fluid mechanics. However, the inability to synoptically measure the horizontal structure of eddies at high resolution—particularly dynamic quantities—during their lifetime remains a significant observational gap, hindering the global validation of these theoretical and numerical studies.

In the ocean, mesoscale eddies form a complex "soup" of currents, yet some eddies exhibit distinct patterns—such as a large diameter, recurrence, long lifetimes, and isolation—that make them ideal candidates for studying their specific dynamics
in greater detail. Most famous examples lie in the western boundaries of oceanic basins: the Gulf Stream rings (Richardson,



1983), the Agulhas rings (Olson and Evans, 1986), the Kuroshio rings (Li et al., 1998), or the Loop Current Eddies (LCEs, Meunier et al., 2018). The latter are large, recurrent features in the Gulf of Mexico, whose evolution has been extensively studied both numerically and observationally since decades (Hamilton et al., 1999). LCEs form at the boundary of the Loop Current, stirring the surrounding water masses as they drift, shaping the hydrography of the Gulf of Mexico (Sheinbaum et al.,

2024). LCEs have been shown to impact pollutant transport (see *e.g.,* Goni et al., 2015). They are also key for heat transport, thus creating strong air-sea exchanges, particularly important in this area for hurricane intensification (Shay et al., 2000).

The ability to track the dynamics of eddies such as LCEs has been made possible through gridded altimetry, which allows for the detection of their spatial extent and evolution over time (Mason et al., 2014). Despite its usefulness in recovering some of the eddies' dynamical properties (*e.g.,* radius, velocity, amplitude, etc.), this approach has large flaws, mainly due to the

fact that the spatiotemporal interpolation often fails to provide full confidence in accurately capturing the horizontal shape of oceanic features (Le Traon et al., 1998) from their birth until their death.

The Surface Water and Ocean Topography (SWOT, Morrow et al., 2019) program, a collaborative effort between CNES and NASA launched in late 2022, marks a pivotal advancement in the measure of mesoscale eddies in the ocean. With its unprecedented spatial resolution and 2D instantaneous coverage, SWOT enables synoptic observation of ocean features with

fine-scale precision. In this study, we demonstrate the ability of SWOT in capturing the full evolution of a mesoscale eddy, focusing on a Loop Current Eddy in the Gulf of Mexico. This highlights the capacity of modern altimetry to bridge the gap between numerical predictions and real-time observations of oceanic vortices.



## 2 Synoptical description of the full Loop Current Eddy life cycle

SWOT's non-interpolated swaths (see full details in appendix A) provide continuous, high-resolution observations of a LCE's
horizontal structure during its whole lifetime (Figs. 1a-r). This allows us to describe in detail its evolution from formation in
early October 2023 to decay and disappearance by the end of March 2024.

In August 2023, a pre-existing large anticyclone ($LCE_0$) was positioned near 27°N, 90°W. Around the same time, the
Loop Current overshot into the Gulf of Mexico, forming a Loop Current Tongue ($LCT_0$, also referred to in the literature as
an "Extended Mode Loop Current"). The latter subsequently merged with $LCE_0$ (Figs. 1a-d), absorbing it in what has been
referred to as a "Reattachment" sequence, a process previously identified in regional numerical modeling studies of the Gulf
of Mexico (Oey et al., 2003) and gridded altimetry. In early October, the Loop Current detached, generating the LCE that is
the focus of this study. This newly formed eddy was then trapped within a train of alternating-polarity eddies (Figs. 1d-f). As it
stirred surrounding vortices of opposite polarities–marked by negative Sea Level Anomaly (SLA) patches–it created a strong
vorticity shield (VS, Valcke and Verron, 1997). By early December, the LCE had settled into a nearly circular shape, its VS
effectively *isolating* it from external influences (Fig. 1g). Note that this LCE is rather small compared to the average size of
LCEs (Meunier et al., 2024), which is usually the case when the Loop current is in Extended mode as observed here.

In December 2023, the LCE undergoes a *destabilization* sequence. Its VS transitions into a rotating train of four cyclones
($C_{1-4}$, see Figs. 1h-l). This kind of structure has been referred to by various names in past studies, including "squared vortex",
"pentapole", "mode-4 vortex", etc. It has long been predicted by numerical simulations and theoretical studies (see *e.g.,* Morel
and Carton, 1994; Chérubin et al., 2006, and our own simulations in Appendix B) and stands as a textbook example of eddy
destabilization. Such vortical structures with intense satellite vortices are typically self-destructive (Carton, 2009). However,
in this case, external conditions and interactions with surrounding eddies prevent the destruction of the mode-4 LCE, leading
to a complex interplay of processes.

In January 2024, a new Loop Current Tongue ($LCT_1$) forms as the Loop Current undergoes another detachment (Fig. 1m).
Unlike in August 2023, when $LCT_0$ merged with a pre-existing eddy ($LCE_0$), this time, satellite cyclones $C_{2,4}$ act as a barrier:
the presence of these vortices prevents the merging of $LCT_1$ with the LCE (Figs. 1o). This directly confirms previous model
and altimetry-based predictions (Le Hénaff et al., 2014) of LCE shedding being blocked by cyclonic activity in the eastern
Gulf of Mexico. Here, we further show that these cyclonic structures are generated by the LCE's own instabilities, and play
a major role in determining its fate. As the system evolves, two of the cyclonic satellites, $C_1$ and $C_2$, are expelled eastward
by $LCT_1$, leaving the LCE with only two remaining companions (Figs. 1n-p). While SWOT does not provide full coverage of
their trajectory, it is likely that $C_1$ and $C_2$ eventually merge. This alters the symmetry of the system, leaving the LCE coupled
with a single cyclone. The result is the formation of a dipole (Figs. 1p).

In mid-February, the dipole begins to drift steadily westward (Figs. 1p-r), guiding the LCE toward the well-known *eddy
graveyard* at the western boundary of the Gulf of Mexico (Biggs, 1992; Tenreiro et al., 2018). The dipole slightly deviate
northward due to the asymmetry of the dipolar structure. As it moves, the LCE gradually loses energy and weakens, as seen
by its loss in SLA amplitude, and as previously demonstrated in Meunier et al. (2020). This westward drift is not driven by





planetary Rossby wave propagation, as traditionally assumed in the literature (see *e.g.,* Chelton et al., 2011b; de Marez et al., 2020b). Instead, it is a direct consequence of the dipolar interaction between the LCE and its remaining cyclone. This challenges the conventional view of how Loop Current Eddies migrate and underscores the importance of mesoscale interactions and

stability properties of the Loop Current Eddies in shaping their trajectories.

We were able to analyze the complete life cycle of a Loop Current Eddy with unprecedented detail. Previous studies relied on coarse altimetry, requiring strong assumptions to reconstruct eddy evolution (see *e.g.,* Hamilton et al., 1999), while high-resolution observations, such as sea surface temperature, provided only partial insights without direct velocity estimates (Hall and Leben, 2016). This observational gap left key dynamical processes—particularly the emergence of high-mode instabili-

ties—hidden from direct detection and often questioned as possible artifacts of altimetric gridding (Le Traon et al., 1998).

More broadly, in the global ocean, classical altimetry studies primarily characterized eddy self-evolution through their *ellipticity*, a proxy for mode-2 deformations affecting velocity estimates (Ioannou et al., 2019). However, they could not resolve more complex structures. With SWOT's high-resolution altimetry, as demonstrated here, we now observe higher-order modes, revealing that eddies are not merely elliptical but can exhibit much sharper gradients than previously thought. This challenges

the long-standing view of eddies as circular structures and suggests that velocity estimates commonly used in global studies (Zhang et al., 2014) may require revision using SWOT's high-resolution data.

The present observation confirms that the mechanisms governing oceanic mesoscale eddies evolution–instability growth, eddy shielding, and dipolar interactions–are fully consistent with theoretical and numerical "vortex studies" dating back to the 1980s (*e.g., Ikeda, 1981; Gent and McWilliams, 1986).* However, unlike past descriptions that often treated these pro-

cesses separately, we now see that these textbook mechanisms are not isolated processes; rather, they are deeply entangled, continuously interacting and shaping the oceanic mesoscale eddies evolution in complex ways.

Being able to observe the horizontal shape of long living eddies with such accuracy is particularly significant for the vortex dynamics community, which has long relied on numerical and theoretical predictions to study mesoscale eddies in the ocean. The lack of direct, high-resolution observations has remained a major limitation, leaving key aspects of vortex evolution

unverified in real-ocean conditions. With SWOT, we can now access unambiguous evidences that the textbook mechanisms of vortex instability and decay hold in the ocean, reinforcing four decades of theoretical and numerical work. These results mark a turning point, opening new perspectives for both theoretical and modeling studies of oceanic eddies and their role in large-scale ocean circulation, but also in helping the near real-time monitoring and interpretation of the vertical structure of those eddies through *e.g.,* autonomuous glider deployment (Meunier et al., 2018).





## 3  Observation of secondary instabilities at the rim of the eddy

Beyond the mesoscale dynamics governing the LCE's evolution, a closer look at its periphery reveals the presence of intense submesoscale instabilities.

Until now, the sharpness of eddy rims has been largely masked by low-resolution altimetric products. Here, the highest-resolution SWOT product (with a 250-m spacing grid) resolves the steep SLA gradients at the edge of the LCE. We present two particular times, where 4 distinct SWOT passes sampled the quasi-totality of the LCE's northern edge, one before its destabilization sequence early December (Figs 2a-c), and another one after the mode-4 structure emerged (Figs. 2d-f). The latest SWOT data processing now provides direct velocity estimates at the eddy's edge, revealing remarkably fine-scale dynamics. Current magnitudes reach up to $\mathcal{O}(1)\,\mathrm{m\,s^{-1}}$, with velocity variations over very short distances, resulting in sharp velocity gradients. These gradients are particularly enhanced between the satellite cyclone and the LCE core in January (Fig. 2e), reinforcing the idea that the primary instability of the LCE further steepened these gradients. This confirms previous studies using numerical modeling discussing the velocity gradient sharpness at LCEs' edges (Hiron et al., 2021).

These conditions are well known to generate submesoscale features through various dynamical instabilities, including barotropic, baroclinic, and centrifugal instabilities (McWilliams, 2019). Additionally, in the case of the LCE described here, a surface mixed layer is present (about 100 m deep, as seen in *in situ data* not shown here), favoring Surface Quasi-Geostrophic (SQG) dynamics. This enhances the development of submesoscale structures (see our SQG simulation reproducing the LCE conditions in Appendix B and *e.g.,* Capet et al., 2008). Here, we observe that the strong frontal gradients at the eddy's rim trigger *secondary instabilities*, a process extensively described in de Marez et al. (2020a). This leads to the formation of intense submesoscale coherent vortices (SCVs) and filaments with high vorticity values ($\zeta/f > 1$), as clearly visible in Figs. 2c,f.

Numerical models have long predicted the formation of such fronts, filaments, and submesoscale coherent vortices at the edges of mesoscale eddies (see for example Fig. 2 of Brannigan et al., 2017). However, observational evidence of such structures has remained scarce. On the one hand, snapshots from high-resolution satellite imagery, such as chlorophyll-a concentration, have only hinted at their presence (Lévy et al., 2018). On the other hand, occasional *in situ* measurements have reported isolated snapshots of submesoscale processes at eddy peripheries (in particular in LCE cases using sea gliders, see Meunier et al., 2019; Pérez et al., 2022). Thus, without synoptic velocity fields, direct observations of these intense vorticity structures were lacking. This raised a fundamental question in numerical oceanography: were these submesoscale features real, or merely numerical artifacts arising from *ad hoc* parameterizations? For the case of this LCE, SWOT provides an unambiguous observational answer. Applied to the global ocean, it will therefore likely reveal the ubiquity of intense submesoscale turbulence around eddies.

Submesoscale secondary instabilities play a crucial role in the long-term decay of eddies, including the LCE observed in this study. These instabilities drive the gradual loss of energy and mass of eddies, as momentum and tracers are redistributed over shorter time scales than those associated with mesoscale processes alone. This mechanism provides a pathway for energy dissipation. Our observations align with previous *in situ* studies showing that LCEs steadily lose amplitude as they drift westward (Meunier et al., 2024). This has important implications for heat transport and air-sea interactions. Eddies like the LCE store and




redistribute large amounts of heat during their lifetime (Meunier et al., 2020), but the mechanisms controlling their heat loss
remain uncertain. Submesoscale processes likely enhance lateral heat release by increasing diffusivity at the eddy periphery
(Gentil et al., 2024). Along with recent studies, this suggests that existing parameterizations of eddy-driven heat fluxes may
need revision and update to account for submesoscale processes.

## 4  Conclusions

This study marks a significant advance in satellite oceanography by providing the first synoptic, high-resolution observation
of an entire mesoscale eddy life cycle. It also illustrates that high-resolution satellite altimetry can resolve submesoscale
instabilities at the edge of an eddy, providing new insights into the small-scale processes that contribute to eddy decay.

While this represents a single case study, the implications are far-reaching. With SWOT's unprecedented resolution, the
oceanographic community can now extend this approach globally, enabling a systematic characterization of mesoscale eddy
life, and submesoscale turbulence across all ocean basins. This is a crucial step toward closing the oceanic energy budget, bridg-
ing the gap between mesoscale eddy dynamics and smaller-scale dissipative processes. Unlike machine learning approaches,
which often provide parameterizations without physical interpretability, SWOT will likely allow us to observe, analyze, and
ultimately derive physics-based parameterizations of small-scale turbulence. This ensures a deeper understanding of ocean
dynamics and paves the way for improved modeling of the Earth's complex systems.



**Figure 1.** Evolution of the Loop Current Eddy (LCE) over its lifetime from 2-km resolution SWOT altimetry. Each panel shows the 1/4°
gridded AVISO Sea Level Anomaly (SLA) as the background color for the indicated date. Superimposed are the 2-km denoised SWOT SLA
measurements along passes within ±3.5 days of the given date. The black contour marks the LCE detection from gridded altimetry, while
thin gray contours represent iso-SLA lines from SWOT passes at 5 cm intervals.



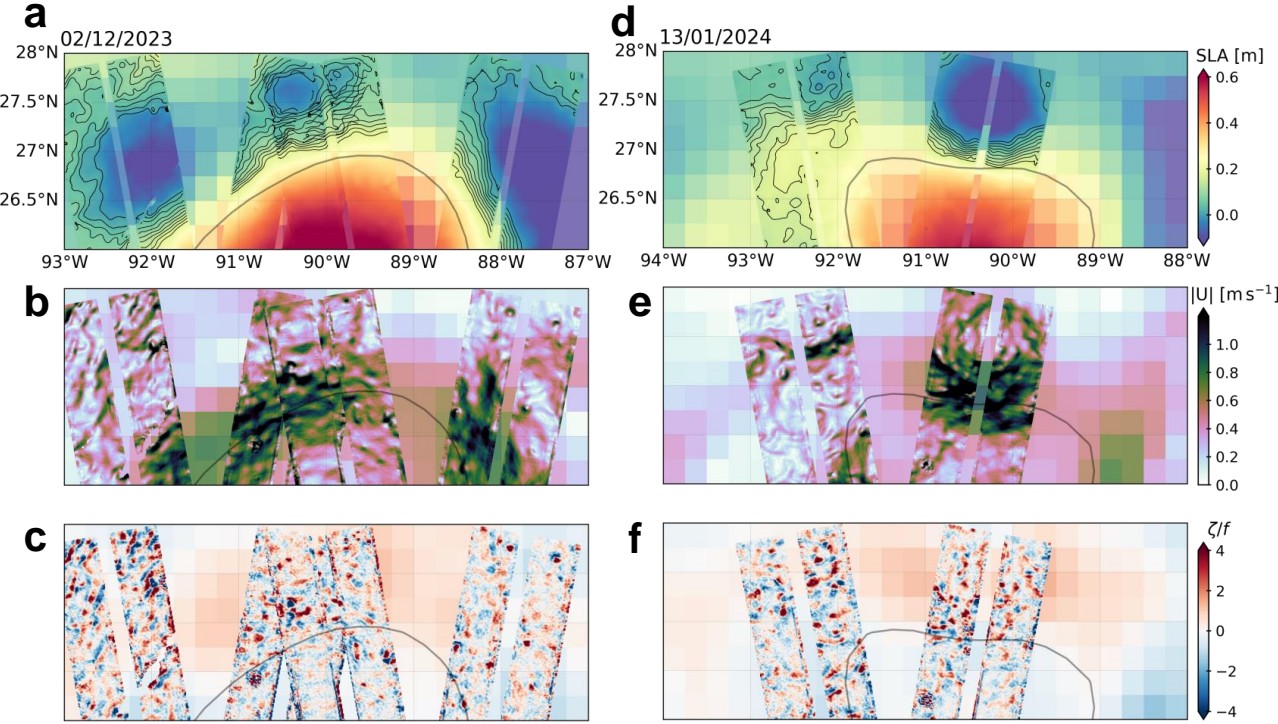

**Figure 2.** Observation of secondary instabilities at the rim of the Loop Current Eddy from 250-m resolution SWOT altimetry. (a,d) Same as Figs. 1g,m, but zoomed in and using the 250-m denoised SWOT SLA product. Black contours indicate iso-SLA lines from SWOT passes at 2 cm intervals. (b,e) Geostrophic current magnitude derived from the denoised SWOT SLA. (c,f) Normalized relative vorticity computed from the geostrophic currents shown in (b,e).

## Appendix A: SWOT altimetry

We leverage newly released SWOT satellite data (see some background in *e.g.,* Morrow et al., 2019), and in particular, we use the latest release (v2.0.1) of Level-3 SWOT data, namely SWOT_L3_SSH 'Basic' (2-km resolution) and 'Unsmoothed' (250-m resolution) products, derived from the L3 SWOT KaRIn Low rate ocean data products provided by NASA/JPL and CNES. The Level-3 processing removes SWOT's systematic errors, and has been extensively validated using other altimeters, numerical models, and *in situ* data, in the global ocean (Dibarboure et al., 2025). These datasets are produced and freely

distributed by the AVISO and DUACS teams as part of the DESMOS Science Team project (AVISO/DUACS, 2023a, b). We use the "denoised" SLA for our analysis (see Dibarboure et al., 2023, for details on the method), as well as the geostrophic velocity derived from it. Note that this process tends to reduce the overall energy, which complicates the interpretation in lower-energy regions due to the noise levels (see *e.g.,* Callies and Wu, 2019). This has no implication in our case, given the



very high energy level of the LCE and its surrounding. However, note that this procedure may lead to spurious velocity patterns at the edge of swaths, thus leading to vorticity artifacts that should not be taken into account for the analysis. Recent studies (Zhang et al., 2024a, b; Verger-Miralles et al., 2024; Du and Jing, 2024; Damerell et al., 2025; Wang et al., 2025; Tchilibou et al., 2025; Carli et al., 2025; Dibarboure et al., 2025; Coadou-Chaventon et al., 2025) and ongoing personal work show the SWOT's ability to resolve small sutructures previously undetected in gridded products. Therefore, although a complete *in situ* validation is not yet available in our study area, we have no doubt that SWOT's resolution is able to capture the dynamics of the LCE and its surrounding.

To track the eddy's lifetime, we identified its center position each week using a gridded altimetric product (not shown). Around this position, we selected the closest SWOT tracks sampled within $\pm 3.5$ days of the detected eddy center date. Since SWOT measurements are instantaneous, this approach ensures spatiotemporal continuity. It also guarantees that, in each panel of Fig. 1, time gaps between passes remain too small to affect our conclusions. For example, SWOT's passes in Fig. 1g are from passes 175, 203, 244, and 272 of SWOT's cycle 7, respectively sampled on 29/11, 30/11, 2/12, and 3/12/2023. This approach ensures, for example, (1) a full spatiotemporal coverage of the $LCT_0$ during the LCE formation (Figs. 1a-d); (2) the fact that the small cyclones $C_{1-4}$ are LCE satellites and not part of another larger cyclones (Fig. 1l); and (3) that the dipole in Figs. 1o-r remains coherent over time, driving the westward drift of the LCE.

**Appendix B: Numerical modeling**

Using a simple numerical setup we reproduce some of the classical results found in earlier "vortex studies" that mirror our observations.

First, we integrate a two-layer Quasi-Geostrophic (QG) model in order to observe the emergence of azimuthal modal structures from the destabilization of a LCE-like eddy. Following the formulation of de Marez and Callies (2025), the problem is non-dimensionalized in space as $x, y \sim R_D$, and in time $t \sim R_D/U$, where $R_D$ and $U$ are the baroclinic deformation radius, and a current magnitude scale, respectively. Also, the aspect ratio between the upper and bottom layers thicknesses, $h_1$ and $h_2$ is $\delta = h_1/h_2$. In this framework, the only parameters that vary are $U$, $R_D$, and $\delta$, and are chosen to be representative of the LCE described here: $U = 0.8\,\mathrm{m\,s^{-1}}$, $R_D = 50\,\mathrm{km}$, and $\delta = 0.25$. Note that the results presented here were not sensitive to these parameters (not shown). The domain size is chosen to be a 800 km wide square, and simulations are ran over a year. Simulations are integrated in either $128 \times 128$ or $512 \times 512$ points grid, and are denominated as "low resolution" or "high resolution" runs, respectively. Timesteps are adjusted to respect the CFL criterion. Small scales are damped through hyperviscosity of order $n = 10$ coefficient $\nu$, and chosen to be the smallest as possible, similarly as in Callies et al. (2016). We chose a doubly periodic domain to eliminate spurious boundary effects, similarly as in classical numerical vortex studies (see all the the literature since McWilliams and Flierl, 1979).

In vortex stability studies, the key aspect is the initialization of the vortex. Numerous analytical models exist, with the simplest being a Gaussian. However, the choice of which quantity follows a Gaussian distribution significantly impacts the results. A Gaussian vorticity profile leads to a very stable vortex, which is not representative of realistic oceanic conditions. In





contrast, a Gaussian streamfunction or velocity profile better reflects oceanic vortices, as it naturally generates a vorticity shield. This shield is crucial for the vortex's evolution and closely represents the conditions observed in our study, as seen in *e.g.,* Fig. 1g. Here, we adopt this approach and initialize the upper-layer streamfunction (equivalent to the SLA) of the model as a
Gaussian-like function

$$\psi_1 = \pm A e^{-\left(\frac{r}{R}\right)^\alpha}, \tag{B1}$$

with $r$ the radial coordinate, $A$ the amplitude of the eddy, $R$ its radius, and $\alpha$ the steepness of the eddy-edges. Note that we also tested a "Mexican hat" function for $\psi_1$ initialization—designed to provide both an SSH and a vorticity shield—and found similar results (not shown). We chose $R = 100\,\mathrm{km}$. Mainly, the simulations show that the presence of the shield leads to the
formation of multiple satellite vortices, with their number increasing as the steepness parameter $\alpha$ increases, with no impact on the number of "coherent structures" from the resolution of the model (Figs. B1a-e). In particular, the case with $\alpha = 5$ shows that eddy can naturally become mode-4 structures, similarly as observed in SWOT data (Fig. 1l).

Second, we integrate a Surface Quasi-Geostrophic (SQG) model, part of the same code as the QG model discussed earlier, following Callies et al. (2016). The SQG model is expressed in physical units, with parameters representative of typical open
ocean conditions: Coriolis parameter $f_0 = 10^{-4}\,s^{-1}$ and Brunt–Väisälä frequency $N_0^2 = 8 \times 10^{-4}\,s^{-2}$. We use the same numerical setup as in the QG model (resolution, viscosity, timestep) and initialize it identically to Fig. B1d,e ($\alpha = 5$, $R = 100\,\mathrm{km}$). In the simulation, a mode-4 structure emerges, indicating that its formation is intrinsic to the eddy's initial shape rather than the physical model. However, we also observe numerous submesoscale structures around it (Fig. B1f), a hallmark of SQG dynamics. This closely resembles the edges of the observed LCE, where many fronts and SCVs are detected.



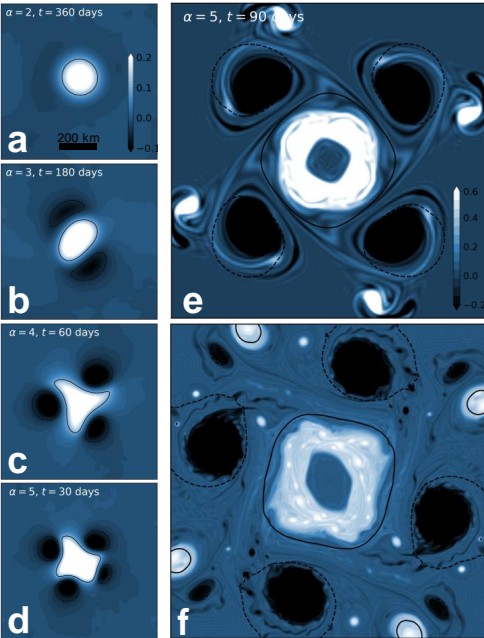

**Figure B1.** Eddy stability numerical simulations. (a–d) Snapshots of the upper-layer streamfunction from low-resolution two-layer Quasi-Geostrophic (QG) simulations, initialized with $\alpha = [2, 3, 4, 5]$. (e) Upper-layer vorticity snapshot from a high-resolution two-layer QG simulation, using the same parameters as in (d); dashed and solid contours show -0.05 and 0.1 streamfunction contours, respectively. (f) Surface vorticity snapshot from a high-resolution Surface Quasi-Geostrophic (SQG) simulation, also using the same parameters as in (d).

*Code and data availability.* SWOT data can be downloaded on AVISO website https://www.aviso.altimetry.fr/en/my-aviso-plus.html. Numerical model is available on github https://github.com/joernc/QGModel.

*Author contributions.* CDM conducted the analysis, performed the simulations, and wrote the manuscript.

*Competing interests.* The author declares no competing interests.

*Acknowledgements.* CDM was supported by a Queen Margrethe II´s and Vigdís Finnbogadóttir's Interdisciplinary Research Centre on
Ocean, Climate and Society (ROCS) postdoctoral fellowship. We thoroughly thank Xavier Carton, Angel Ruiz-Angulo, and Thomas Meunier for their valuable feedback and insightful comments on the manuscript. The author acknowledges the many scientists who have contributed to the long history of research in the Gulf of Mexico. Special recognition is also given to the Gulf of Mexico itself—not only for its rich





and complex dynamics but also for the significance of its name, which serves as a crucial reminder of its true identity and must be protected whatever the cost.



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
