# Peer review of "Synoptic observation of full mesoscale eddy lifetime and its secondary instabilities in the Gulf of Mexico"

_EGUsphere, 2025_

## Referee Comment (RC2)

**Poseidon**

Meunier et al 2018.

**Figure 2**

**Sn**apshots of ADT (m) from 01 April to 01 November 2016. The 0.7-m ADT isopleth is materialized by a black contour line. The red dots stand for the centroid of the 0.7-m ADT isopleth. The grey dots represent the location of the Argo profiles available during this time lapse. A list of the latter is available in Table **A1**. ADT = Absolute Dynamic Topography.

[Figure]

**Kraken from low res altimetry (https://www.aoml.noaa.gov/phod/dhos/altimetry.php)**

[Figure]

Farily ciruclar with no clear appearance of cyclones near its rim. Cyclones were present surrounding it as part of the detachment from the Loop Current, and also the large cyclone present in the western basin which alter interacted with Kraken as it approached the western boundary.

See more on the coherence of this eddy in Beron-Vera et al 2018.

**LCE in this study with low res altimetry**

[Figure]

[Figure]

[Figure]

[Figure]

Very active mesoscale field present when this eddy was formed

Cyclonic circulations surrounded it even before it was "born" (4cyclones were surrounding it before it detached from the LC, september/october 2023)

4 cyclones in squared vortex configuration visible in 23 and 30 December 2023 (fig 1j,k in paper)

**LCE in this study with low res altimetry, lifecycle**

[Figure]

Notice dipole for February 17 and 24, 2024 (fig 1p,q)

Lifecycle includes nearly splitting in two (March), and apparently merging in August with the LCE that was formed in March (last panel is an animation). Hence its lifetime lasted at least from December 2023 to August 2024.

---

## Author Comment (AC1)

RC1: 'Comment on egusphere-2025-1592', Anonymous Referee #1, 23 Apr 2025

Review of "Synoptic observation of full mesoscale eddy lifetime and its secondary instabilities in the Gulf of Mexico" by Charly de Marez.

This is a concise, well written paper summarising observations of a loop current eddy's life cycle in the Gulf of Mexico using high resolution altimetry from the relatively recently launched SWOT satellite. Direct observation of key dynamical processes such as eddy shielding, high-mode instabilities, and dipolar interactions were made possible by the resolution and coverage of the SWOT data. Although short and highly focused, this paper provides significant insight into processes which likely affect eddies throughout the global ocean. Some minor improvements to the text and figures are needed, but once completed I would recommend this paper for publication.

We thank the reviewer for their careful reading and insightful comments, which helped improve the clarity and depth of the manuscript, our answers to all the questions raised in the review can be found below written in blue.

**Comments on figures**

All figures: the Ocean Science style guide says panel labels should be enclosed in brackets on the figure. We modified the panel labels accordingly.

Figure 1:

Fig 1l looks a lot like figure eleven in this journal's typeface – you could miss the letter l out of your panel labels and just go j, k, m. We modified the labels accordingly

I can see why you've chosen to display this the way you have – you get all of it on one figure, and the days you want to emphasise are larger. However, I think skipping between the larger and smaller panels does disrupt the flow for the reader because now you don't just start at top left, work across the row, go down one row, work from left to right, etc. Given that you currently only have two figures in the whole paper, I suggest you consider stretching this out - perhaps three figures each with six panels of equal size, for example? We agree with the reviewer that this sub-panel design could disrupt the flow. However, we choose to keep only two figures in the whole manuscript to stick to the OS Letters format.

Are you sure the colour scale is colourblind-accessible? And personally, I find diverging colour scales which aren't centered at zero rather confusing.

We thank the reviewer for this important check. We passed the figure in https://www.color-blindness.com/coblis-color-blindness-simulator/ to check for color blind accessibility and it

seems that is ok. If the reviewer think it is not, we would be grateful they indicate it to us. Concerning the diverging color, we chose to present the result this way to emphasize the satellites/LCEs and their opposite vorticities.

I think the AVISO SLA is maybe a bit too pale.  For example, I really can't see $LCT_1$ on panel m, it just looks white.  You could make the AVISO SLA quite a bit less pale while still maintaining a good contrast between it and the SWOT SLA. We understand the referee, and we believe that in panel m we cannot see the LCT1 because it is in the "white range" of the colormap. We choose to let this tone for the AVISO data to not mislead readers on what part of the data is interpolated or not, and also acknowledge that the colormapping is modified for the AVISO SLA in the caption.

Panels b, c, d, e, h, I, j, k, n, o, p, q have text above them which is far too small to read – I had to zoom in to 200% to see it was their dates.  Since they're not all regularly spaced in time, the reader needs to be able to see the dates.  We increased the date fontsize accordingly. These panels also lack axes labels, and panels n and p have arrows which are not explained in the caption nor explicitly mentioned in the main text. We added description of the axes label in the caption and the discuss the arrows in the text at lines 75,79.

The labels $LCE_0$, $LCT_0$, LCE, VS, $C_1$, $C_2$, $C_3$, $C_4$ and $LCT_1$ are not explained in the caption, nor have they been mentioned in the main text at the point where you first refer to this figure.  I see that there's a lot of explanation in the main text of these labels, and it's understandable that you don't want to put all of this in the caption, but perhaps you could say something like "The labels $LCE_0$, $LCT_0$, LCE, VS, $C_1$, $C_2$, $C_3$, $C_4$ and $LCT_1$ will be discussed in the main text." We thank the reviewer for the suggestion and we added this mention to the caption.

You don't need the colourbar three times on one figure, but you do need to label it. We added a label to the colorbar, and removed the duplicate.

Figure 2:

On figure 1, you said "The black contour marks the LCE detection from gridded altimetry, while thin gray contours represent iso-SLA lines from SWOT passes at 5 cm intervals".  Here on figure 2 you say "Black contours indicate iso-SLA lines from SWOT passes at 2 cm intervals", but it looks like you still also have the black contour which is the LCE detection from gridded altimetry, even though this isn't mentioned in the caption. We indeed forgot to mention the LCE detection contour in the caption. We corrected it accordingly.

Panels a and d – is this exactly the same colour scale as in figure 1?  It worth be worth either saying so in the caption, or altering the colourbar so it has the same tick marks as in figure 1, which will make it more obvious that it's the same scale.  In any case, the AVISO SLA is clearly not as pale here as on figure 1, and it would be better if the two figures were consistent. The

colorbar were not the same, so we modified the colormapping of Fig. 2 for consistency. We now acknowledge this in the caption of Fig. 2.

Re the black contours which indicate iso-SLA lines from SWOT passes at 2 cm intervals – it looks like you're only showing these in certain areas, i.e., outside the contour marking the LCE detection from gridded altimetry, but this isn't stated in the caption. It is exact, we only show contours in the range 0<SLA<0.2m. We added this information in the caption.

Panels b and e show "Geostrophic current magnitude derived from the denoised SWOT SLA", but you clearly have data outside the SWOT swaths. Ditto panels c and f. The background color is derived from AVISO SLA, we added this precision in the caption.

In figure 1 your panel labels went left to right and then down to the next row. In figure 2 your panel labels go down the first column and then down the second column. It's easier for the reader if you do them the same way in all figures. Ditto figure B1. We modified Figs. 2,B1 accordingly.

Panels b to f have no axes labels or visible tick marks. I think there are grid lines on all panels but they're so faint they're very difficult to see. Please make them like figure 1's grid lines, and add axes labels. You don't need axes labels on all panels but you do need x-axis labels on the bottom row and y-axis labels on the left-hand column. We modified the figure accordingly

Figure B1:

The colourbars and length scale are very difficult to see, even zoomed in. Please put them outside the panels.

The text at the top of each panel is also quite small and hard to read.

The dashed and solid contours on panels e and f are difficult to see – maybe use a contrasting colour?

We modified the figure to gain visibility on all the points mentioned by the referee.

**Textual comments**

Line 52: "This newly formed eddy was then trapped within a train of alternating-polarity eddies (Figs. 1d-f)." To my eyes, Fig 1d doesn't look that different to panels b or c, so it's not clear to me at what point you would start describing something as "trapped within a train of alternating-polarity eddies". Indeed, however, starting from Fig. 1d, a clear positive SLA signal remains at the LCE's position, while negative SLA anomalies begin to develop both to the west and east of it.

This marks the emergence of the alternating-polarity pattern typical of a vortex train, which is not yet visible in earlier panels. We clarified the sentence accordingly.

Line 128: Seagliders is a specific brand name, the generic term is ocean gliders. We modified the text accordingly

Line 168: sutructures We modified the text accordingly

Line 171: which gridded altimetric product?  Even if you don't want to show it, you could name it. We now mention the name of the product in the text.

Line 173:  SWOT measurements are not instantaneous.  It takes time for the satellite to orbit around the globe.  Perhaps you meant that the measurements are near-instantaneous compared to timescales of interest? We agree with the reviewer, and modified the text accordingly using the reviewer's formulation.

Line 174: "time gaps between passes remain too small to affect our conclusions."  Could you say a little more here about the timescales of interest?  It is clear from figure 1 that changes do occur on weekly timescales.

It is true that Figure 1 shows changes on a weekly timescale, but the typical timescale of eddy evolution due to baroclinic instability is on the order of weeks to months, as illustrated in Appendix Figure B1. Given this, the ±3.5-day window used to combine SWOT passes is well below the relevant dynamical timescale and does not blur the key features of interest. We clarified this at lines 185-194.

Line 188: "simulations are ran over a year."  This is bad grammar, please correct. We corrected the grammar

Line 190:  "Timesteps are adjusted to respect the CFL criterion."  You don't say what this criterion is, or what CFL stands for, or provide a reference. We have clarified what the CFL criterion is, spelled out the acronym, and added a reference in the revised manuscript, see lines 205-206.

Line 191: "the smallest as possible" – either "the smallest possible" or "as small as possible".  Plus "similarly as in" is bad grammar.  Perhaps just "as small as possible (Callies et al., 2016)." We modified the text accordingly

Line 192:  delete "similarly". We modified the text accordingly

Line 204:  "We chose R = 100km" because? Because it is the same size as the LCE studied here. Note that this parameter does not have an impact on the conclusions we rise. We clarified this in the text.

---

## Author Comment (AC2)

RC2: 'Comment on egusphere-2025-1592', Anonymous Referee #2, 28 May 2025

This paper describes the sea surface height and vorticity structure at the high resolution obtained by SWOT for the case of one Loop Current Eddy, from the moment of its formation to part of its evolution as it drifts westwards. It is certainly nice to see how the promise of SWOT actually looks like in this region after all these years of research using numerical models and sparce data to get an idea of what these measurements may be able to resolve.

We thank the reviewer for their careful reading and insightful comments, which helped improve the clarity and depth of the manuscript, our answers to all the questions raised in the review can be found below written in blue.
* * *
**General comments based on the reviewer's concerns**

I-

With some of their comments, the reviewer was questioning the claim that features like dipolar interactions, vorticity shielding, and the full eddy life cycle have already been observed prior to SWOT. They argue that many of these features appear — at least partially — in low-resolution, gridded altimetry products, and that LCEs and their dynamics was already fully observable before SWOT measurements. We thank the reviewer for these valuable comments. We acknowledge that some large-scale features of the LCE, such as dipolar structures or approximate trajectories, are visible in low-resolution, gridded altimetry products. However, we would like to emphasize a key distinction: these gridded datasets are interpolated products, derived from along-track altimetry, which do not directly resolve mesoscale or submesoscale structures at eddy-periphery scales. Also, these products are interpolated using Gaussian-based space-time reconstruction methods, which can smooth out fine-scale features and, in some cases, introduce artifacts that resemble eddies, rather than capturing real, instantaneous structures.

In contrast, SWOT provides direct two-dimensional observations of sea level anomaly at high spatial resolution, without requiring interpolation. This allows us to observe with confidence the emergence and spatial structure of the cyclonic companions to the LCE, including their relative position and coherent shape—features that are only partially or ambiguously visible in interpolated altimetry.

Our intention is not to dismiss previous observations, but rather to clarify that SWOT offers the first unambiguous, high-resolution observation of these companion cyclones evolving alongside a Loop Current Eddy throughout its lifetime.

We have revised the text accordingly to moderate the claims and to better highlight this distinction, see some answers to the following specific comments below:

*Maybe tone down a bit "this has not been previously characterized but in theoretical or numerical models". At least dipolar interactions (see figures later on) can be observed even with the low res altimetry products and other observations prior to SWOT.*

We modified the abstract sentence as follows : "SWOT allows for the direct observation of key dynamical processes—such as eddy shielding, high-mode instabilities, and dipolar interactions---that were previously only partially resolved in interpolated altimetry products and mainly described in theoretical or numerical studies."

*L 54. Note that the vorticity shield is not really shown but rather inferred from the shape of the eddy. I am not convinced the cyclonic vorticity ring around the LCE is not observed with low res altimetry, how do you determine that eddy shielding can* only *be inferred from SWOT data, as claimed in the abstract?*

*The eddy instability process can* only *be inferred by SWOT, or in other words, the low res altimetry data would not be able to provide hints of it. Particularly given that the cyclones that appear are of approximately the same scales as the LCE. At least in the eddy in question, the low res does show hints of those cyclones*

*L 94. Given all the above, "The present observation confirms…. " and "fully consistent with theoretical* and numerical "vortex studies"" seem to strong statements.*

We modified our statement at the end of the section 2 to clarify this:
"We were able to analyze the complete life cycle of a Loop Current Eddy with unprecedented spatial detail. Previous studies relied on low-resolution, gridded altimetry products that required spatio-temporal interpolation and strong assumptions to reconstruct eddy evolution \citep[see {\it e.g.,}][]{hamilton1999loop}, while high-resolution observations, such as sea surface temperature, provided only partial insights without direct velocity estimates \citep[][]{hall2016observational}. This observational gap limited the ability to clearly identify key dynamical features—such the emergence of high-mode instabilities—which, in interpolated altimetric fields, can be smoothed or distorted and occasionally misrepresented \citep[][]{le1998improved}."

*L 145 I would tone this down too, since the eddy was still "alive" at the end of the study period. I did continue having quite an interesting evolution that can be traced by low res altimetry until at least July 2024 (see attached file)." We modified the sentence accordingly :" This study marks a significant advance in satellite oceanography by providing the first synoptic, high-resolution observation of the full development and mature evolution of a mesoscale eddy, captured continuously over several months."*

II-

The reviewer pointed out that the comparison between the SWOT SLA field and the idealized simulations were sometimes misleading, or not documented enough. Specifically, the reviewer raises important questions about (1) whether the observed cyclones truly originate from instability, (2) whether the quadrupole structure is consistent with theory or potentially pre-existing, (3) how well the idealized simulations compare in terms of timescales and dynamics, (4) the role of external conditions in modifying the evolution, and (5) whether similar structures could be seen in low-resolution altimetry. In light of this, we have revised the manuscript to clarify three key points:
(1) the role of the simulations as qualitative tools rather than predictive models;
(2) the recognition that some of the observed cyclones may not form purely by instability but their configuration is still consistent with mode selection;
(3) external interactions, often neglected in idealized studies, are critical in modulating the eddy's evolution.
We have also softened language in the text to avoid overstating the generality of our conclusions and to better distinguish what is observed directly versus inferred from theoretical context. Below, we gathered and respond to each of the reviewer's specific comments about this specific thematic.

*The numerical results that back up what is observed are certainly a very nice complement, but are barely described. I would find a much more convincing case that what is observed with SWOT is that particular eddy instability with a bit more detailed comparison. For example, do the 4 cyclones so formed around the LCE evolve in aproximately the same time scales (the observations show that 4-mode state lasts just a couple of weeks, how do these timescales compare in terms of, say, number of inertial periods in the simulations and in the LCE)? How many inertial periods does the eddy last from the moment the instabilities appear and its destruction, and how does that time compare to the period the LCE was followed? Do you see such things such as migration away from the eddy (C1 and C2) or merging to evolve into the dipolar structure that is associated to the westward migration of the eddy?*

We agree that a more direct comparison between the observations and the idealized simulations would be valuable. However, it is important to clarify that our use of the simulations is primarily qualitative, aiming to provide a mechanistic framework rather than a one-to-one match. The real ocean environment—in the Gulf of Mexico or elsewhere—is significantly more complex than our idealized double-periodic setup, and the fate of the LCE and its companion cyclones is largely influenced by interactions with external features (e.g., surrounding eddies and background flows), which are absent in the simulations. Most previous idealized simulation studies have overlooked the role of external influences, which we believe limits the realism of their conclusions; we highlighted this point in the context of vortex mergers in a past study of ours:

*de Marez, C., Carton, X., L'Hégaret, P., Meunier, T., Stegner, A., Le Vu, B., & Morvan, M. (2020) Oceanic vortex mergers are not isolated but influenced by the β-effect and surrounding eddies, Scientific Reports,*

and the same principle applies here to the eddy destabilization process. Therefore, while the instability appears to select a dominant mode-4 structure both in the model and in the observations, a quantitative comparison of timescales (e.g., duration of the quadrupole state or number of inertial periods until decay) is difficult to make and may be misleading. Also, the migration of cyclonic satellites away from the eddy, is only determined by external influences in our case. We now clarify this point in the revised text at lines 232:

"We emphasize that the simulations presented here are intended to illustrate the dominant instability mechanism qualitatively. The subsequent evolution and timescales observed in the real ocean are highly influenced by external interactions absent from the idealized model, making direct comparisons hazardous."

*L57-80. I myself find it hard to tell if C1-4 where formed by eddy destabilization, given that cyclonic structures were already present surrounding the LCE0 formation, and may be precursors of those 4 cyclones associated to the "squared vortex", and they last a very short period of time in that layout. Would be nice to see how the evolution fo the theoretical case shown in Appendix B looks in comparison as those cyclonic eddies evolve, see my general comment on that in the beginning of this review.*

We understand the reviewer's concern, and we acknowledge that we have no more direct proof of this cyclonic formation than the SWOT SLA snapshots shown in Fig. 1. Therefore, we propose to say that we do not ensure that the four cyclones (C1–C4) are strictly generated by baroclinic instability. Rather, we say that the coherent quadrupole configuration observed in SLA is consistent with a mode-4 structure maintained and amplified by the instability. This distinction is important. The cyclones may have pre-existed as weak anomalies, but their growth into a quasi-symmetric and persistent configuration strongly suggests the selection and imprint of an underlying instability mechanism. This interpretation is supported by the similarity of the pattern to the theoretical mode-4 evolution shown in Appendix B. We have clarified this distinction in the revised manuscript at lines 61-62:

"While the companion cyclones may not all be generated \textit{de novo} by the instability, their coherent configuration and evolution suggest that the LCE instability reinforces a dominant mode-4 structure."

*Another thing to point out is that, even though not as nicely, the low res altimetry does show that layout (see attached file). This may be not so surprising given that the cyclones that evolve from the instabilities are comparable in size to the original LCE according to the simulations. I say this cause there are examples of LCEs that have evolved in a relatively quiet mesoscale background (Poseidon, Meunier et al. 2018; Kraken, Beron-Vera et al. 2018, see attached file) which lasted a long time (months) without showing such strong deformations that could as clearly be attributed to eddy instabilities such as those in figure B1. Given that Poseidon was particularly large and energetic and does not seem to interact with that many other mesoscale structures present in the gulf, wouldn't it have evolved instabilities with such cyclones and be destroyed quicker? How*

*comparable are the time scales in the simulation from the appearance of the cyclones and the destruction of the eddy to the lifetimes of Kraken and Poseidon?*

We thank the reviewer for pointing out the presence of similar patterns in gridded low-resolution altimetry. Indeed, large-scale features such as dipolar structures or quadrupoles may appear in interpolated SLA maps. However, we stress here that these gridded fields result from spatio-temporal interpolation, often using Gaussian filters, which can smooth or distort the true structure of mesoscale features and occasionally generate eddy-like artifacts, particularly at the scale of satellite cyclones. In contrast, SWOT data offers direct, two-dimensional, high-resolution observations, allowing for a confident identification of the spatial arrangement and coherence of the companion cyclones.

Regarding Poseidon and Kraken, we fully agree that not all LCEs undergo this instability-driven quadrupole evolution, and this may depend on their initial structure, background strain, and mesoscale environment. Our goal is not to generalize the mode-4 evolution as universal, but rather to show that in this case, the observed evolution matches the theoretical signature of a mode-4 instability, as also illustrated in Appendix B.

*The description of the SSH evolution of figure 1 is* proof *that the LCE destabilized as expected from theory. At the most I would say this particular case invites to think of this instability process. See our general comment.*

As discussed above, we have moderated our conclusions to reflect that the observation is consistent with the expected behavior of a mode-4 instability, and we have pointed out the importance of external influences, rather than being definitive proof of a unique mechanism.

*The eddy did not get destroyed by its own instability because "external conditions and interactions with surrounding eddies prevent the destruction of the mode-4 LCE, leading to a complex interplay of processes." Can you describe more clearly how you unequivocally arrive at this conclusion?*

We acknowledge that this statement was too strong as originally phrased. What we observe is that the eddy, after transitioning into a mode-4 configuration, does not follow the self-destruction pathway often seen in isolated simulations. Instead, it persists as a coherent structure while interacting with surrounding eddies. Based on the deviation from the idealized evolution, we infer that these external interactions are responsible for preventing the typical decay, but we agree that this is not demonstrated unequivocally. We have revised the sentence at lines 65-68 to reflect that this is an interpretation based on the comparison between simulations and observations, not a definitive conclusion.
* * *
Nevertheless, I consider that given the information provided many statements and conclusions need to be toned down, particularly in Section 2 and the Conclusions, or in some cases

elaborating a little more on how the conclusions are reached is necessary. I provide specific examples below.

Another general comment is that it would be nice to have include more references that back up or provide more guidance to the reader in the Introduction (below I point to specific cases), particularly with respect to the "theoretical and numerical understanding of mesoscale eddy dynamic… offering insights into eddy behavior." so as to have a better idea of the theoretical and conceptual framework, and the details they entail. Elaborating a little on that would also be helpful.

The numerical results that back up what is observed are certainly a very nice complement, but are barely described. I would find a much more convincing case that what is observed with SWOT is that particular eddy instability with a bit more detailed comparison. For example, do the 4 cyclones so formed around the LCE evolve in aproximately the same time scales (the observations show that 4-mode state lasts just a couple of weeks, how do these timescales compare in terms of, say, number of inertial periods in the simulations and in the LCE)? How many inertial periods does the eddy last from the moment the instabilities appear and its destruction, and how does that time compare to the period the LCE was followed? Do you see such things such as migration away from the eddy (C1 and C2) or merging to evolve into the dipolar structure that is associated to the westward migration of the eddy? See our general comment above

Section 3 I find no large objections to since indeed this is where the SWOT resolution certainly shows something that was not easily observed before around LCEs. It would be nice though if something is said on how much we can trust the vorticity obtained from SLA at those small scales and high Rossby numbers (i.e. geostrophic currents from SLA slopes), see for example Tranchant et al 2025. Although I do believe this is the first time I see in an article the observations of SWOT on a LCE, there certainly is more literature where SWOT has been used to reveal submesoscale features surrounding mesoscale fields with some in depth analysis on how much certain processes are represented (e.g. Agulhas retroflection in Coadou-Chaventon 2025). These are all very recent and may not have been available to the author when he submitted, but I think some of those results may help make a stronger case for what is being stated here with only a few snapshots of the SWOT derived fields. Including such references at least in the ending statement of the Conclusions would certainly help to make a more informative and stronger case for SWOT's potential to solve all that is mentioned there.

We added a paragraph in the method section to discuss this specific point and we now cite 3 relevant papers including the ones suggested by the reviewer, see lines 177-184.

**Detailed comments**

Abstract

Maybe tone down a  bit "this has not been previously characterized but in theoretical or numerical models". At least dipolar interactions (see figures later on) can be observed even with the low res altimetry products and other observations prior to SWOT. See our comment about this point above.

L 20. some citations would be helpful. We added a reference to the vortex study literature.

L 45. Previous sentence talks of all LCEs, this sentence implies it talks of a specific one. Maybe repharse to say this will be possible to be done to LCEs present in that period of time. We rephrased accordingly.

L 51. Can you provide reference with an example of a reatachament observed with gridded altimetry reattachment? We added reference to Manta et al. 2023, which discuss reattachment sequences from the detection of LCEs in gridded altimetry.

L 54. Note that the vorticity shield is not really shown but rather inferred from the shape of the eddy. I am not convinced the cyclonic vorticity ring around the LCE is not observed with low res altimetry, how do you determine that eddy shielding can *only* be inferred from SWOT data, as claimed in the abstract? See our comment about this point above.

L 55. Please provide reference defending that the difference in sizes of LCEs depends on if they are formed with and extended vs no extended Loop Current. In any case, maybe this info doesn't matter much since nothing has been stated on why the size of the eddy would matter for the rest that is being discussed. We agree with the reviewer and removed this rather qualitative statement.

L57-80. I myself find it hard to tell if C1-4 where formed by eddy destabilization, given that cyclonic structures were already present surrounding the LCE0 formation, and may be precursors of those 4 cyclones associated to the "squared vortex", and they last a very short period of time in that layout. Would be nice to see how the evolution fo the theoretical case shown in Appendix B looks in comparison as those cyclonic eddies evolve, see my general comment on that in the beginning of this review. See our comment about this point above.

Another thing to point out is that, even though not as nicely, the low res altimetry does show that layout (see attached file). This may be not so surprising given that the cyclones that evolve from the instabilities are comparable in size to the original LCE according to the simulations.  I say this cause there are examples of LCEs that have evolved in a relatively quiet mesoscale background (Poseidon, Meunier et al. 2018; Kraken, Beron-Vera et al. 2018, see attached file)  which lasted a long time (months) without showing such strong deformations that could as clearly be attributed to eddy instabilities such as those in figure B1. Given that Poseidon was particularly large and energetic and does not seem to interact with that many other mesoscale structures present in the gulf, wouldn't it have evolved instabilities with such  cyclones and be destroyed quicker? How comparable are the time scales in the simulation from the appearance

of the cyclones and the destruction of the eddy to the lifetimes of Kraken and Poseidon? See our comment about this point above.

Don't get me wrong, I would more than happy be convinced, but as of now I don't find evidence that support the following conclusions:

1. The eddy instability process can *only* be inferred by SWOT, or in other words, the low res altimetry data would not be able to provide hints of it. Particularly given that the cyclones that appear are of approximately the same scales as the LCE.  At least in the eddy in question, the low res does show hints of those cyclones. See our comment above.

2. The description of the SSH evolution of figure 1 is *proof* that the LCE destabilized as expected from theory. At the most I would say this particular case invites to think of this instability process. See our comment above.

3. The eddy did not get destroyed by its own instability because "external conditions and interactions with surrounding eddies prevent the destruction of the mode-4 LCE, leading to a complex interplay of processes." Can you describe more clearly how you unequivocably arrive at this conclusion?  See our comment above.

4. "The westward drift is not driven by planetary Rossby wave propagation, as traditionally assumed… instead, it is a direct consequence of the dipolar interaction". How is that so clearly determined from these images? could it be both? Note that the dipole is clearly seen in the low res altimetry (images provided in attached file), while no such dipole is present for Poseidon nor Kraken, and they also migrated westwards.  From how all of this is phrased, it seems to imply that  LCEs migrate westwards because they are in fact dipoles, and those dipoles only exist because of the evolution of the cyclones that appear due to the eddy instability. But this is certainly not always the case. We agree that the phrasing in the original version may have overstated the generality of the mechanism. Our intention was to describe the behavior observed in this particular case, where the westward drift of the LCE seems associated with the dipolar interaction visible in the SLA fields. We acknowledge that this is not the only mechanism by which Loop Current Eddies can migrate westward, and that planetary Rossby wave dynamics or other processes may dominate in other cases. We have revised the paragraph in the manuscript to clarify that our statement applies specifically to the evolution of the LCE analyzed in this study, see lines 83-88.

L 94. Given all the above, "The present observation confirms…. " and "fully consistent with theoretical and numerical "vortex studies"" seem to strong statements. See our general comment.

L 97. Maybe add what a vorticity shield refers to, if I understand it right it is a ring of opposite vorticity (cyclonic) surrounding the eddy. We clarified this sentence accordingly.

L 100 " unambigous evidence…." at least from the evidence provided in this study, seem too strong a statement here. We removed the adjective unambiguous to tone down this sentence.

L 145  I would tone this down too,  since the eddy was still "alive" at the end of the study period. I did continue having quite an interesting evolution that can be traced by low res altimetry until at least July 2024 (see attached file). See our comment above.

Figure 1. Would be nice if labels for C1-C4 in panels m - r are included (merging also, maybe as C1+C2 or something like that) to better follow the sequence described in the text. We agree with the reviewer and we modified the figure accordingly.

REFERENCES

Beron-Vera, F.J., Olascoaga, M.J., Wang, Y. et al. Enduring Lagrangian coherence of a Loop Current ring assessed using independent observations. Sci Rep 8, 11275 (2018). https://doi.org/10.1038/s41598-018-29582-5

Coadou-Chaventon, S., Swart, S., Novelli, G., & Speich, S. (2025). Resolving sharper fronts of the Agulhas Current Retroflection using SWOT altimetry. Geophysical Research Letters, 52, e2025GL115203. https://doi.org/10.1029/2025GL115203

Meunier, T., Pallas-Sanz, E., Tenreiro, M., Portela, E., Ochoa, J., Ruiz-Angulo, A., and Cusí, S.: The vertical structure of a Loop Current Eddy, J. Geophys. Res. C: Oceans, 123, 6070–6090, https://doi.org/10.1029/2018JC013801, 2018.

Tranchant, Y.-T., Legresy, B., Foppert, A., Pena-Molino, B., & Phillips, H. E. (2025). Swot reveals fine-scale balanced motions and dispersion properties in the Antarctic circumpolar current. Authorea Preprints.

---

## Referee Report (RR1)

Final revision

I thank the author for the discussion over my concerns and subsequent clarification in the text. I just found a couple more places where some rephrasing would be appropriate that I did not mention specifically in my original review.

Given we agreed that the LCE was still alive long after March 2024, and haow the conclusions were changed, please rephrase these sections in the ABSTRACT:

Yet, their full life cycle has never been observed synoptically at high resolution. In this study, we use novel SWOT satellite altimetry data to present the first synoptic characterization of a Loop Current Eddy's life cycle in the Gulf of Mexico, from formation to dissipation.

…

These findings not only validate decades of vortex theory….

Here some other places which for the same reason would be good to do some rephrasing:

L42. full evolution of a mesoscale eddy…

L48. between its formation in early October 2023 to its decay and disappearance by the end of March 2024.

Finally these few places with minor comments

L59. I would get rid of the statement "Note that this LCE is rather small compared to the average size of LCEs (Meunier et al., 2024)". On the one hand because Poseidon that is the one studied in Meunier et al 2024 does not represent an average size (it was unusually large), and on the other given that we agreed size does not particularly matter for the discussion presented.

L106-107. I would tone down by writing something like "The present observations show evidence that mechanisms…. which are consistent with theoretical and numerical…"